# Electrospun Konjac Glucomannan/Polyvinyl Alcohol Long Polymeric Filaments Incorporated with Tea Polyphenols for Food Preservations

**DOI:** 10.3390/foods13020284

**Published:** 2024-01-16

**Authors:** Liying Huang, Ronglin Liao, Nitong Bu, Di Zhang, Jie Pang, Ruojun Mu

**Affiliations:** College of Food Science, Fujian Agriculture and Forestry University, Fuzhou 350002, China13950487886@163.com (R.L.); zdfst@fafu.edu.cn (D.Z.); pang3721941@fafu.edu.cn (J.P.)

**Keywords:** konjac glucomannan, polyvinyl alcohol, tea polyphenols, electrospinning

## Abstract

In this study, nanofiber films were prepared by electrospinning technology with polyvinyl alcohol (PVA) and konjac glucomannan (KGM) as raw materials. Tea polyphenols (TPs) were incorporated in the above matrix, which increased physicochemical (thermal and mechanical characteristics) and antibacterial properties of the nanofiber films. The release behavior of phenolic compounds from PVA/KGM-TPs nanofiber films was determined in different food simulants; antioxidant and antibacterial activity of the films were also evaluated. The results showed that the addition of KGM increased the physical and chemical properties of the films. The tensile strength (TS) and elongation at break (EB) increased from 5.40 ± 0.33 to 10.62 ± 0.34 and from 7.24 ± 0.32 to 18.10 ± 0.91, respectively. PVA/KGM-TPs nanofiber films performed controlled release of TPs, with final release of 49.17% in 3% acetic acid, 43.6% in 10% ethanol, and 59.42% in 95% ethanol. The nanofiber films showed good antioxidation properties, with the free radical scavenging rate increasing from 1.33% to 25.61%, and good antibacterial properties with inhibition zones against *E. coli* and *S. aureus* of 24.33 ± 0.47 mm and 34.33 ± 0.94 mm, respectively. In addition, the as-prepared films showed significant preservation performance for raw bananas at 25 °C.

## 1. Introduction

Food packaging plays a crucial role in preserving and ensuring the quality and safety of food products. It aims to extend the shelf life of foods by isolating various physical, chemical, and microbiological hazards. To achieve those objectives, many advanced technologies such as casting [1], melt blending [2], polymerization [3], electrospinning [4], and materials such as petroleum-based plastic polymers [5], biodegradable polymers [6], and paper [7] have been employed in development of food packaging. Nanofiber films, derived from nanotechnology, have emerged as a promising option for food packaging due to their unique properties and versatile applications [8,9,10,11]. Electrospinning is one of the most popular methods of building these types of structures. Fabrication of long polymeric filaments (LPFs), also called nanofibers, is a cheap, versatile, promising, well-established, and scalable process [12]. LPFs have special properties, such as high surface-area-to-volume ratio, high porosity, high encapsulation efficiency, and controllable morphology [13]. The specific physical, chemical, and mechanical properties of LPFs rely on the characteristics of polymers and the parameters of the electrospinning process [14,15]. Viscosity, conductivity, surface charge, and surface tension were identified as the main factors affecting uniform and fine fiber formation [11]. Based on the various functions of LPFs, they has been widely applied in many scientific areas, such as adhesion of neurons [16], cell growth [17], transportation of bioactive compounds [18], sensor development [19], enzyme immobilization [20], and creation of biodegradable food containers [12].

Due to the broad applications of LFPs, both synthetic and natural polymers have been explored to obtain this nanoscale structures [21]. Nowadays, ideal green solvent systems and natural resources are used to achieve the goals of “Green Chemistry” [22]. Natural polymers offer unique advantages over synthetic polymers, such as biocompatibility, biodegradability, chemical stability, and lack of toxins, with appropriate mechanical properties. Konjac glucomannan (KGM) is one of the natural polymers that can be applied to produce LPFs [23]. It is a renewable natural polymer resource, like cellulose and starch. KGM is a polysaccharide with extremely long polymer chains (200~2000 kDa), hence is capable of holding water [24] and forming films [25]. However, disadvantages of biopolymers applied in the process of electrospinning are obvious. (1) Unstable generation of LPFs due to ultra-high viscosity of polymer solutions [25], and (2) brittle structure and poor mechanical properties of LPFs [26]. Those limitations decrease the application of KGM in the preparation of LPFs via electrospinning technology [23]. Therefore, crosslinked networks with two or more polymers are one of the commonly used strategies to provide new materials with better performance than those with single networks.

It has long been investigated whether double networks and triple networks can significantly increase mechanics of soft materials like hydrogels [27,28]. Our previous study also proved that double networks of KGM and polyethylene oxide (PEO) LPFs performed better mechanically than those with single networks [29]. Polyvinyl alcohol (PVA) is one of the most commonly used synthetic polymers in the electrospinning process [30,31]. It is a biodegradable, water-soluble, and biocompatible polymer with hydrophilic chains [32,33,34,35]. However, similar limitations as KGM exist when preparing LPFs with PVA single networks. Therefore, many scientists focused on the improvement of mechanical properties of PVA LPFs with other biopolymers. It has been reported that chitosan [36], gelatin [37], starch [38], and tara gum [39] are all promising natural resources to form double networks with PVA during the process of spinning technologies. In double network gels/films, the tensile strength at break of PVA was significantly increased from 20.71 to 44.74 MPa. Meanwhile, composite nanofiber films could be a good substrate for loading natural active products, which would impart new properties to the nanofiber films, such as antioxidant and antibacterial properties.

In this work, we present the strategy for construction of LPF films by using KGM and PVA via electrospinning technology. Physical and mechanical properties of LPF films, such as water content (WC), water vapor permeability (WVP), water contact angle (WCA), oxygen permeability (OP), tensile strength (TS), swelling degree, Fourier transform infrared spectroscopy (FT-IR), X-ray diffraction (XRD), and thermogravimetric analysis (TGA) were measured. The morphology, chemical structures, thermal stability, antibacterial, and antioxidant activities of the films were observed. Our strategy offers the following advantages: (1) KGM is a biodegradable polysaccharide that endows PVA-KGM nanofiber films with degradable properties, meeting the requirements of modern green packaging. (2) PVA-KGM nanofiber films have good mechanical strength and flexibility. (3) The addition of KGM enhances the hydrophilicity of the fiber film, which is beneficial for the moisturizing and preservation of food. (4) PVA-KGM nanofiber film can extend the shelf life of food through excellent oxygen barrier performance. The release behavior of TPs in films was investigated in 3% acetic acid, 10% ethanol, and 95% ethanol. In addition, the objective of these experiments was to evaluate the effect of the films on the preservation of raw bananas. Our method is to measure the weight loss, firmness, titratable acidity, and total soluble solids of postharvest bananas after 15 days, which we can more fully evaluate the effect of PVA/KGM-TPs in food preservation.

## 2. Materials and Methods

### 2.1. Materials

Tea Polyphenols (TPs), 2,2-diphenyl-1-picrylhydrazyl (DPPH), Konjac glucomannan (KGM) and Polyvinyl alcohol (PVA) were acquired from Aladdin Chemical Reagent Co., Ltd. (Los Angels, CA, USA), *Staphylococcus aureus* (*S. aureus*) and *Escherichia coli* (*E. coli*) were purchased from Keborui Biotechnology Co., Ltd. (Shanghai, China). Additional analytical-grade chemical reagents were obtained from Sino-pharm Group Chemical Reagent Co. Ltd. (Shanghai, China).

### 2.2. Preparation of Nanofiber Films

TPs were added to KGM solution (1%, *w*/*v*) and stirred for 12 h to obtain KGM solution containing 5 mg/mL TPs. PVA solution (6%, *w*/*v*) was prepared by dissolving 6 g PVA in 100 mL deionized water and stirring the mixture using magnetic stirring for 2 h at 90 °C in a water bath. Subsequently, the 6% PVA solution was cooled to 40 °C. A series volume of KGM-TPs solutions was added to PVA (6%, *w*/*v*) solution (100 mL) to obtain five solutions, labeled P1, P2, P3, P4, and P5, with KGM-TPs concentrations (*w*/*v*) of 0%, 0.5%, 1%, 2%, and 3%, respectively. All solutions were stirred at 40 °C for 12 h. Each solution was dispensed into a 20 mL plastic syringe fitted with a needle (Model: 20G). Finally, electrospinning process was applied to prepare nanofiber films. The parameters of electrospinning process were set as: 9.5 kV spinning voltage, 1 mL/h feeding speed, and 15 cm receiving distance.

### 2.3. Morphology and Size Distribution of the LPFs

Morphology of nanofiber films was observed by a Hitachi S-4800 scanning electron microscope (SEM) (Tokyo, Japan) following the previous method [40]. The specimens were coated in an argon atmosphere with roughly 20 nm of gold–palladium using a gold sputter module contained within a high vacuum evaporator. The SEM was powered by a 5 kV accelerating voltage. The average diameter and the diameter distribution of the ultrafine LPFs were evaluated by ImageJ using the micrographs randomly selected from 100 LPFs [41].

### 2.4. FTIR Analysis

The KBr pellet method was used to perform FTIR to determine interactions between the components. A Nicolet-6700 spectrometer (Wausau, WI, USA) was used to obtain the FTIR spectra. The wavenumber range was 4000–500 cm^−1^ and each measurement required an average of 32 scans at 4 cm^−1^.

### 2.5. XRD Analysis

The crystal structure of the nanofibrous films was observed by XRD. An X’Pert Pro-diffractometer (PA Analytical B.V., Breda, The Netherlands) was a Cu Kα radiation source with an operating voltage of 40 kV, 35 mA, and the diffraction range was 5–60° (2θ) and the scanning rate was 5° min^−1^.

### 2.6. Thermal Analysis

TGA was performed on a TA Q500 instrument (TA Instruments, Newcastle, DE, USA) with a heating range of 50 to 600 °C under a continuous flow of dry N_2_ gas. DSC was performed on a TA Q500 instrument (TA Instruments, Newcastle, DE, USA) with a heating range from 50 to 600 °C. The temperature rise rate of TGA and DSC was 10 °C min^−1^.

### 2.7. Mechanical Properties

The test was carried out according to the previous methods [42]. The nanofiber films were cut into 50 mm long and 10 mm broad rectangular strips. The thickness was determined by an IP65 screw micrometer (DITRON, Chengdu, China). The tensile strength (TS) and elongation at break (EB) of films were measured by an EZ-SX universal tensile tester (Shimadzu, Kyoto, Japan).

### 2.8. Water Content (WC)

The water content (WC) of the nanofiber films was evaluated according to the previous literature [43]. Film samples were weighed ($W_{o}$) and then dried at 105 °C for 24 h before being weighed again ($W_{i}$). WC was calculated as the percentage of initial film weight loss after drying. In order to ensure the precision of the experiment, three WC measurements were made for each type of nanofiber film. The calculation of WC was based on the formula as follows:$$WC=\frac{W_{o}-W_{i}}{W_{o}}.$$

### 2.9. Water Vapor Permeability (WVP)

Water vapor permeability (WVP) of nanofiber films was evaluated based on the previous method [39]. The bottle containing 3 g calcium chloride (CaCl_2_) was sealed with a nanofiber film. The bottle was placed in a dryer with relative humidity of 75% (RH), and the weight was tested after 96 h. WVP was calculated as follows:$$WVP=\frac{X\times ∆W}{∆t\times A\times ∆P},$$
where the weight change (ΔW) after 96 h, sample interval time (Δt), the sample thickness (X) in millimeters, the sample area (A) in square meters, and the partial vapor pressure (ΔP) are represented by symbols in the formula.

### 2.10. Oxygen Permeability (OP)

The OP was evaluated according to the previous methodology with some modifications [44]. The bottle was filled with 1 g activated carbon, 1.5 g sodium chloride, and 0.5 g iron powder. Then, the bottle was covered with the prepared film. After weighing the bottle, it was stored in a saturated barium chloride solution (90% RH) dryer. After 60 h of measuring the total weight, the OP was calculated by the following formula:$$OP=\frac{∆M}{S\times t},$$
where ΔM is the weight change after 60 h, S is square meters and t is equilibration time.

### 2.11. Water Contact Angles (WCA)

The water contact angles (WCA) of the films were measured using a Contact Angle Meter (OCA 20, Data Physics, Filderstadt, Germany). WCA was defined as the angle between the tangent line and the drop baseline. The distilled water of 10 μL per drop was precisely deposited on the surface of the nanofiber film using a precision syringe.

### 2.12. Swelling Degree

In the swelling test, nanofiber films measuring 1.5 cm × 1.5 cm square were weighed before ($W_{0}$) and after ($W_{t}$) immersion in PBS (pH = 7.4) buffer solution for 24 h [45]. The following formula was used to calculate the swelling ratio: (1)$$swelling\;degree\;(\%)=\frac{W_{0}-W_{t}}{W_{0}}\times 100.$$

### 2.13. Antioxidant Activity

The nanofiber films’ antioxidant capabilities were evaluated by measuring their DPPH radical scavenging activity using the previous approach [23,46]. All experiments were conducted three times. A particular amount of DPPH was dissolved in ethanol at a concentration of 0.04 mg/mL. Samples (30 mg) were added to 3 mL of DPPH solution, immediately shaken, and then stored at room temperature in the dark for 3 h. The solution without samples was recorded as the control under the same conditions. The DPPH radical scavenging activity was determined as follows:$$DPPH\;radical\;scavenging\;activity\left(\%\right)=\frac{A_{0}-A_{i}}{A_{0}}\times 100,$$
where $A_{0}$ represented absorbance of blank, $A_{i}$ represented absorbance of sample.

### 2.14. Antibacterial Activity

To determine the antibacterial activity of nanofiber membranes, samples with a diameter of 20 mm were sterilized with ultraviolet irradiation for 20 min. The bacteria suspensions (10^5^ CFU/mL) were uniformly spread on agar plates. The sterilized samples were placed in the LB agar center in Petri dishes incubated at 37 °C for 24 h. The antibacterial activity was determined through measuring the inhibition zone.

### 2.15. Release of TPs

To observe release characteristics of TPs from nanofiber films, three kinds of food simulants (95% ethanol, 10% ethanol and 3% acetic acid) were selected, representing fatty food, alcoholic food, and acidic food, respectively. A weighed film sample (40 mg) was dipped into 10 mL of each food simulant and shaken well at 100 rpm. 5 mL solution was taken at different times (20 s, 10 min, 20 min, 30 min, 1 h, 2 h, 4 h, 6 h, 12 h, 24 h, 48 h, 72 h and 96 h), and fresh equivalent solution was added [47]. The absorbance of solution was measured through a Shimadzu 1800 ultraviolet visible spectrophotometer (Japan). The standard curves of TPs in fatty food, alcoholic food, and acidic food simulants were y = 32.665x + 0.0642 (R^2^ = 0.9998), y = 30.867x − 0.0362 (R^2^ = 0.9997), and y = 14.525x + 0.0311 (R^2^ = 0.9994), respectively. The standard curve was used to calculate the amount of tea polyphenols released from the film into the food simulant.
$$Cumulative\;release\;\left(\%\right)=\frac{V_{1}C_{n}+V_{2\;}\sum C_{n-1}}{M_{0}}\times 100,$$
where $M_{o}$ represents the mass of TPs in the sample; $V_{1}$ and $V_{2}$ represent the total volume of the release medium and the extraction volume of the release medium at the given time, respectively; and $C_{n}$ and $C_{n-1}$ are the concentrations of TPs in the n and (n−1) releases of the medium, respectively.

### 2.16. Preservation Experiment of Bananas

All freshly picked bananas need to be pre-treated. The surfaces of the bananas were cleaned, and the bananas with intact surfaces and no mechanical damage were selected for the experiment. All bananas were divided into four groups: uncovered, KGM covered, PVA covered, and P3 covered. During the banana preservation process, the raw banana was placed in a transparent acrylic box and the nanofiber film was covered on top of the box (Figure 1d). When it was time for the test, the bananas were taken out onto a black background panel for photos. After the photo was taken, the banana was placed back in the box and covered with a nanofiber film immediately. All bananas were stored at room temperature (25 °C).

#### 2.16.1. Weight Loss and Firmness Determination

The initial weight of banana samples was explicated as *W_0_*. The weight of stored bananas was expressed as *W_t_*. The weight of stored bananas was measured at 0, 1, 3, 5, 8, 12, and 15 days. The determination of mass loss was as follows: $$Weight\;loss\left(\%\right)=\frac{{W_{0}-W}_{t}}{W_{0}}\times 100$$

Firmness of banana was measured using a Texture Analyzer. The tested banana sample was punctured 5 mm in depth with a speed of 15.0 mm/min. The banana firmness was recorded as Newton (N). The firmness of stored banana was measured at 0, 1, 3, 5, 8, 12, and 15 days.

#### 2.16.2. Titratable Acidity Determination

Titratable acidity of bananas was determined by a titration method. Certain bananas were peeled and blended with 100 mL of water. The resultant mixture was filtered through filter paper and centrifugation at 6000 rpm for 15 min. Three drops of 0.1% phenolphthalein indicator were added to the 5 mL supernatant of sample. 0.1 M NaOH solution was then added drop-wise to the solution until the color of the solution changed to pink, which indicated the reaction between base and acids in the bananas. The percentage of malic acid in the fresh bananas is the acidity of the bananas. The titratable acidity of bananas stored for 0, 1, 3, 5, 8, 12, and 15 days was measured.

#### 2.16.3. Total Soluble Solids (TSS)

The total soluble solid contents of the bananas were measured by a digital refractometer (ATAGO PAL-1, Japan). Bananas (1 g) were ground and completely homogenized with 10 mL of distilled water [48]. A drop of banana liquid was dripped on the sample pool of the refractometer to detect the total soluble solids. The TSS contents of stored banana was measured at 0, 1, 3, 5, 8, 12, and 15 days.

### 2.17. Statistical Analysis

Each experiment was repeated at least three times. The experimental data were analyzed using SPSS version 22.0. Statistical significance was determined at *p* < 0.05.

## 3. Results and Discussion

### 3.1. Micro Morphologies of Nanofiber Films

SEM images, nanofiber diameter distribution, and mean nanofiber diameter of PVA nanofiber films with varying KGM content are presented in Figure 2. Uniform LPFs with an average diameter of (152.67 ± 39.94 nm) were obtained via electrospinning of PVA solutions. When adding KGM solution, the morphology of ultrafine LPFs slightly changed, and the average diameter of LPFs increased from 152.67 ± 39.94 to 312.36 ± 279.69 nm. The molecular interactions between KGM and PVA led to the aggregation of PVA molecules into clusters. In addition, under the action of an electric field, the KGM was stretched and pulled to increase the diameter of the LPFs. Because of the elevated surface tension, the stretching of the jet became challenging, leading to the development of rough surface topographies in nanofiber films, and thus mutual adhesion. Therefore, PVA LPFs containing 1% KGM had the best morphology, continuous LPFs, and uniform diameter distribution.

### 3.2. Structure of Nanofiber Films

The primary stretching vibration in the nanofibril film’s FT-IR spectra (Figure 3a) was identified as the -OH absorption peak at 3380 cm^−1^. The mechanical properties of the nanofibril films might be impacted. The C-H and C-O stretching vibrations were represented by the infrared peaks at 2935 and 1027 cm^−1^, respectively, whereas the mannose unit stretching vibrations were represented by the peaks at 879 and 807 cm^−1^. The absorption peak located at 1645 cm^−1^ was attributed to hydrogen bonding inside the molecule. These outcomes matched those of our earlier publications [42]. O-H stretching in the hydroxyl group (3420 cm^−1^), C-H symmetric and asymmetric stretching (2935 cm^−1^), and C-O stretching in the alcohol group (1095 cm^−1^) were identified as characteristic bands of PVA [35].

The FT-IR spectrum showed that, with the addition of pure KGM, the -OH of nanofiber films moved from 3420 to 3450 cm^−1^, and the C-H and C-O peaks were strengthened at 2935 and 1645 cm^−1^. These findings demonstrated that there was a hydrogen bonding between the pure KGM and PVA molecules, and that the hydrogen bonding became stronger. Therefore, hydrogen bond interactions could contribute to good compatibility between polysaccharide and PVA components and lead to better physicochemical and mechanical properties [38].

The effect of KGM on the crystal structure of PVA was studied by measuring the XRD patterns of PVA films with and without KGM (Figure 3b). The characteristic peak of PVA was sharp in the 2θ = 19.8°. KGM is an amorphous material that exhibits major characteristic peaks at 2θ = 20.3° [23]. The addition of KGM increased the intensity of the characteristic peak of PVA nanofiber films and added a shift towards the right, with a weaker peak appearing at 22.4°. This was because the addition of KGM changed the conformation of the PVA molecular chain, thus affecting the arrangement of the crystal structure.

### 3.3. Thermal Stability of Nanofiber Films

Thermal properties of nanofiber films (P1, P2, P3, P4, and P5) were evaluated using TGA (Figure 3c and Table 1), DTG (Figure 3d) and DSC (Figure 3e). TGA images showed that T_−5%_ values (critical significant weight loss of 5%) were higher than 200 °C. Compared with PVA (P1), T_−5%_ values of P2, P3, P4, and P5 nanofiber films were all decreased. To confirm the interactions between the two polymers, DTG values were obtained from the TGA (Figure 3d). It could be observed from the derivative curve that the thermal degradation of PVA takes place in two steps. When the side groups that create water and other volatile chemicals such as ketones and acetaldehyde were eliminated, the pure PVA film reached its original thermal breakdown temperature of 288 °C [49]. The second degradation occurred at 430 °C, which was related to the decomposition of the polymeric backbone chains [39]. Moreover, the T_max_ values of P2, P3, P4, and P5 were all lower than 288 °C. PVA molecular chains were less likely to undergo thermal decomposition reactions, possibly because of the protective effect of KGM’s amorphous structure [50].

PVA is a polymer with a melting temperature ranging between 200 and 240 °C [32]. In the DSC diagram (Figure 3e), the melting peak of PVA nanofiber films appeared at 226 °C. The addition of KGM caused the melting peak of the nanofiber films to shift to the right, indicating that it affected the order of the PVA molecular chain and increased its melting temperature. Furthermore, the PVA film had a peak value at 289 °C, and this peak value shifted to the left after adding KGM. This was because the addition of KGM affected the crystal structure of the PVA molecular chain, resulting in changes in its thermal properties.

**Figure 3 foods-13-00284-f003:**
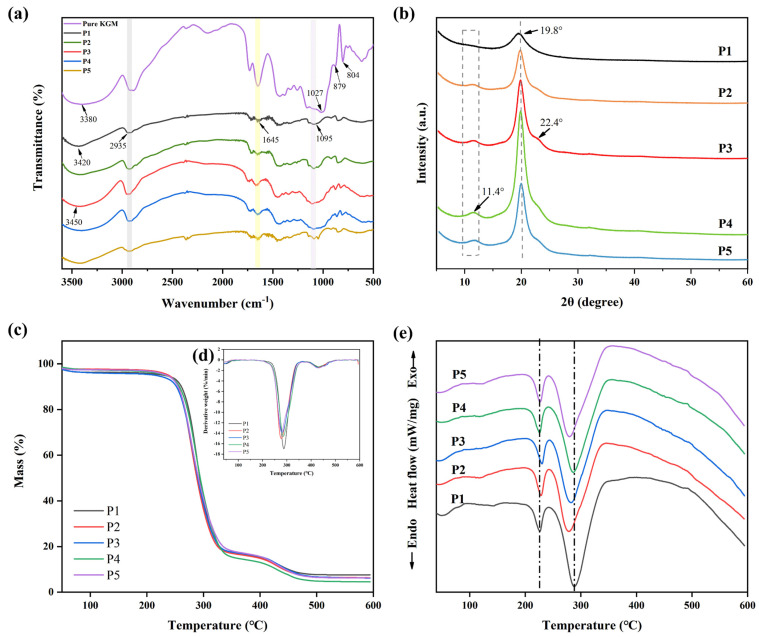
(**a**) FT-IR spectra of P1, P2, P3, P4, and P5 nanofiber films without TPs, (**b**) XRD diffraction patterns of P1, P2, P3, P4, and P5 nanofiber films without TPs, (**c**) TGA curves of P1, P2, P3, P4, and P5 nanofiber films without TPs, (**d**) DTG curves of P1, P2, P3, P4, and P5 nanofiber films without TPs, and (**e**) DSC diagrams of P1, P2, P3, P4, and P5 nanofiber films without TPs.

### 3.4. Physical Properties of Nanofiber Films

To evaluate the properties of PVA/KGM nanofiber films, several factors were analyzed, including tensile strength (TS), elongation at break (EB), water content (WC), water vapor permeability (WVP), water contact angle (WCA), and swelling degree (Figure 2, Table 1 and Table 2). These properties are important for understanding the performance and potential applications of films. By examining the changes in these factors with the addition of KGM, we could gain insight into the effect of this component on the properties of the films. The TS and EB of films were exhibited in Table 1. Films exhibited increased TS (from 5.40 ± 0.33 to 10.62 ± 0.34) and EB (from 7.24 ± 0.32 to 18.10 ± 0.91) compared to pure PVA films. Furthermore, PVA/KGM films possessed a more intricate structure and composition than PVA film. Polysaccharide can increase the crosslinking and intermolecular force of the film, so as to improve the stability and tensile resistance of the fiber film [51]. However, excessive addition of KGM led to increased complexity and aggregation of the internal structure of nanofiber films, resulting in a reduction of mechanical properties.

To assess the permeability of nanofiber films, it is crucial to quantify their WC and WVP [52]. The WC and WVP of films both showed a gradual increase from P1 to P5 (Table 2). Specifically, the WC increased from 3.98 ± 0.37 to 7.76 ± 0.32, while the WVP increased from 3.81 ± 0.08 to 9.61 ± 0.18. KGM is a polysaccharide substance with good water solubility and hygroscopicity. Therefore, when the KGM content increases, it can absorb and retain more water, resulting in an increase in the water content of the nanofiber films. At the same time, the hygroscopicity of KGM may also lead to an increase in the permeability of the film to water vapor, so the WVP will also increase.

In order to prolong the shelf life of food products, food packaging requires an oxygen barrier to slow down the oxidation process. Low oxygen permeability can inhibit anaerobic respiration, thereby achieving the goal of prolonging the freshness of food items. The oxygen permeability (OP) of nanofiber films was depicted in the Table 2. Compared to pure PVA nanofiber films, the OP of nanofiber films containing KGM was reduced (from 3.68 ± 0.15 to 2.72 ± 0.30). This reduction is attributed to the polysaccharide film’s tightly knit and orderly hydrogen bond network structure, which provides an effective oxygen barrier. When the KGM content exceeds 1%, it can increase the aggregation of KGM within the nanofiber films, leading to an increase in OP. These results were consistent with the SEM images of the nanofiber films.

Water contact angle (WCA) is used to estimate water resistance using basic wettability to analyze the hydrophilicity or hydrophobicity of packaging materials. The data in Figure 2 showed that the contact angle of LPFs was in an interval ranging from 29.86 ° to 50.95 ° (*p* < 0.05). Furthermore, it was seen that these values were decreased with the addition of KGM. Lower WCA usually has higher surface hydrophilic properties [53]. Based on this evidence, it was inferred that films synthesized with PVA and KGM had a higher propensity to bind with water. Hydrogen bonds between hydroxyl groups were found to form both intra- and intermolecularly when polysaccharides were bound to polymers such as PVA [38]. With the increase of KGM, hydroxyl functional groups on the surface of film increased. This characteristic increased the hydrophilicity of the polymer by promoting molecular interactions between water and the hydrogen bridge-provided polymer.

Water absorption capacities of the prepared films are investigated by determining their swelling behaviors [37]. Equilibrium swelling ratios were listed in Table 2. P1, P2, P3, P4, and P5 showed excellent water absorption capacity. The high-water absorption capacity was attributed to their main components being highly hydrophilic PVA and KGM. Compared with pure PVA films, the swelling capacity of P2, P3, P4, and P5 decreased slightly from 318 ± 8.68 to 298 ± 7.26. KGM and PVA chains formed complexes that make the structure within the fiber film tighter. This structure limited the absorption and diffusion of water molecules into the film, reducing its swelling capacity.

### 3.5. Antioxidant Property, Antibacterial Property, and Release of TPs in Nanofiber Films

Antioxidant activity of the prepared film is of great significance in its application. DPPH free radical scavenging assay is an evaluation method for the determination of antioxidant activity. After 3 h treatment, the scavenging rates of control, P1, P2, P3, P4, and P5 samples were 1.33%, 1.68%, 14.11%, 18.73%, 20.83%, and 25.61%, respectively. It has been claimed that the high amount of phenolic hydroxyl groups in TPs’ structure may explain their antioxidant activity by effectively donating hydrogen to free radicals and then blocking the chain reaction [54]. In Figure 4a, the capacity for scavenging DPPH free radicals was expressed as TPs, and it was clearly seen that the control and P1 samples showed almost no scavenging capacity. However, the scavenging capacity significantly increased in the presence of TPs, and such changes were closely related to the TP concentration. In P5, the antioxidant activity increased almost two-fold compared to P2.

The inhibition zone images and the diameters of inhibition zones of P1, P2, P3, P4, and P5 are shown in Figure 4. From the inhibition zone diameters, the control P1 did not exhibit any inhibitory effect on *E. coli* or *S. aureus*. However, with the increase in concentration of TPs, the inhibition zone diameter of all tested strains significantly increased. Growth inhibition for *E. coli* and *S. aureus* was 24.33 ± 0.47 mm and 34.33 ± 0.94 mm, respectively. Previous studies had shown that TPs inhibit the growth of a wide variety of bacteria, and were more effective against Gram-positive bacteria than Gram-negative bacteria [55]. In particular, P3, P4, and P5 films showed significant inhibition against *S. aureus*. This was because of the difference in cell wall structure between Gram-positive and Gram-negative microorganisms. Lei et al. made similar observations in pectin/KGM films containing TPs [56]. These results indicated that PVA/KGM/TP films could be used as antibacterial packaging materials in the food industry to extend the shelf life of food products.

Release tests provide information about the affinity between the active substance and the food. The migration of TPs in antibacterial packaging film is related to the type and nature of the food. However, food is a complex system, so it is difficult to study the migration of TPs in food. Thus, the most appropriate active substance can be selected for each food product [57,58]. The release patterns of tea polyphenols (TPs) from 10% ethanol, 95% ethanol, and 3% acetic acid were investigated (Figure 5). In Figure 5, the orange line represents the direct release of TPs, the blue line represents the PVA nanofiber films loaded with TPs, and the green line represents the PVA nanofiber films loaded with KGM-TPs (P3). In all three liquids, the concentration of TPs in the three substrates exhibited a rapid increase at the initial stage, followed by a slow rise after 24 h, and eventually attained equilibrium (0–96 h, 4 days). The release rate of TPs was observed to be the fastest in 10% ethanol, the slowest in 95% ethanol, and moderate in 3% acetic acid. At 96 h, the release rate of TPs in 10% ethanol, a hydrophilic food simulant, was 59.42%, while it was 43.6% in 95% ethanol, a lipophilic food simulant, and 49.17% in 3% acetic acid, an acidic food simulant. These results indicated that the type of liquid food simulant had an impact on the release rate of TPs from the nanofiber films. The low polarity of ethanol facilitated the release of TPs from the nanofiber films. As the concentration of ethanol increased, the internal structure of the nanofiber films became denser [59], and the hydrophilicity of TPs decreased, resulting in a slower release rate. In acidic environments, polyphenolic compounds underwent protonation [60]. This made them more hydrophilic, which increased the rate of TPs release, but not as significantly as in 10% ethanol. PVA/KGM nanofiber films had a more complex structure and composition than PVA nanofiber films. KGM, as a protective material, could enhance the stability and structural strength of the nanofiber films, effectively reducing the release rate of TPs.

### 3.6. Preservation of Banana

It is one of the most difficult duties to correctly preserve bananas effectively in the process of production, transportation, and sale. As a result, bananas were chosen as representative fruits to evaluate the preservation performance of films [61]. Bananas were divided into four groups, and their freshness was monitored for 15 days at room temperature. After 15 days of storage (Figure 6a), the bananas in the control group and those treated with KGM and PVA films almost perished. The banana peel exhibited a partial yellow look after 15 days of storage, indicating that the fresh-keeping performance of bananas had greatly improved in P3 film as a result of the inclusion of TPs. This was due to a decrease in the rate of chlorophyll breakdown caused by reducing the rate of respiration and moisture escape from the banana [62].

Weight loss is a characteristic indicator of banana maturity. The weight loss of all samples gradually increases with storage time from Figure 6b. After 15 days of storage, the calculated mass losses for the first three groups of samples were 35.72%, 35.12%, and 35.57% (from left to right), respectively. The calculated weight loss of the last group of bananas was 27.30%, and the weight loss of bananas was significantly reduced. This indicated the effectiveness of P3 film on postharvest bananas. In addition, the utilization of nanofiber films could play a role in blocking the penetration of moisture, carbon dioxide and oxygen, thereby, decreasing the water loss, metabolic reaction and respiration process [63]. This was also why the weight loss of bananas covered with KGM and PVA film was slightly lower than that of the control. TPs are used as free radical scavengers to reduce oxidative substances in contact with bananas, which cause metabolic reactions in fruits. Therefore, P3 film can effectively reduce the weight loss of bananas.

Firmness is a significant factor in fresh bananas for a variety of reasons, including economic losses caused by reduced transportability, storage time, and postharvest shelf-life. During the ripening process, the peel texture and the cell walls, intracellular materials, and cell structures are degraded, which decrease banana firmness [62]. Herein, it was found that the firmness of all bananas gradually declined as the storage period increased (Figure 6c). The firmness of the first three groups of samples significantly decreased after 15 days of storage (0.29, 0.40, and 0.35 N, respectively), which was closely related to weight loss. The firmness of bananas preserved in P3 decreased slowly during storage (from 14.56 to 1.37 N). This was due to the fact that the covering may restrict metabolism and gas exchange, resulting in the slowing of the banana softening. These results indicated that P3 film could be used to delay the ripening process of bananas and slow down water loss.

Malic acid is the main organic acid in bananas, and it is another predictor of banana storage life. Figure 6d shows the titratable acidity (TA) of uncovered and covered bananas after 15 days of storage. Overall, the total acidity of all bananas gradually decreased throughout the storage period. TA in the blank group decreased from the initial 0.37% to 0.05%, indicating that a large number of organic acids were consumed. However, the TA of P3-treated bananas was 0.11%, indicating that P3 film could reduce the consumption of organic acids and delay the ripening of bananas. This was because the P3 film could cause the fruit’s respiration rate to decrease, thus hindering the consumption of titratable acidity [64]. Therefore, we found that P3 film effectively delayed the aging of bananas by analyzing the changes in weight, firmness, and acidity of bananas during storage.

Total soluble solids content is one of the important indicators to judge the quality of bananas after harvest. The effect of storage time on the TSS percentage of covered and uncovered bananas is shown Figure 6e. Initially, all bananas had a 2.18% TSS (*w*/*w*). The percentage of TSS content in all covered and untreated bananas increased during storage. This may be due to the conversion of starch into other small-molecule sugars. In addition, bananas that were not covered, PVA-covered, and KGM-covered had significantly higher TSS percentages after 8 days of storage. On the other hand, P3-covered bananas had lower TSS. Due to starch decomposition and fruit drying during storage, the TSS of uncovered and nonactive-factor-covered fruits may increase. These results were similar to the reported articles [65].

## 4. Conclusions

In summary, we have prepared a novel nanofiber film with PVA and KGM by using electrospinning technology. To make the best application of the films, TPs were incorporated in the above matrix. The PVA/KGM-TPs nanofiber films presented increased physicochemical (thermal and mechanical characteristics), antioxidative, and antibacterial properties. Most importantly, physical and chemical characterization results revealed that strong molecular interactions occurred between PVA and KGM molecular chains in films. Dependent on the physical entanglements between polymers and nanofabrication from electrospinning technology, the as-prepared nanofiber films are good candidates for food packaging. Our further investigations proved that small molecules (TPs) could be control-released from LPFs, and films performed significant preservation of raw bananas at 25 °C.

## Figures and Tables

**Figure 1 foods-13-00284-f001:**
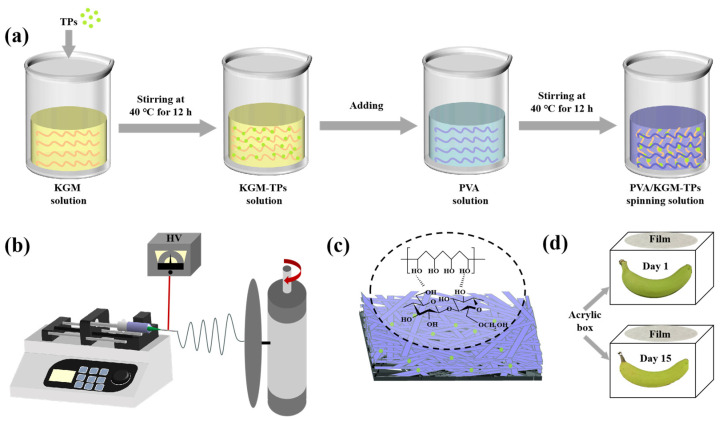
(**a**) Preparation of spinning solution, (**b**) Schematic illustration of PVA/KGM-TPs nanofibrous films, (**c**) PVA/KGM-TPs nanofibrous films, and (**d**) Preservation of the bananas.

**Figure 2 foods-13-00284-f002:**
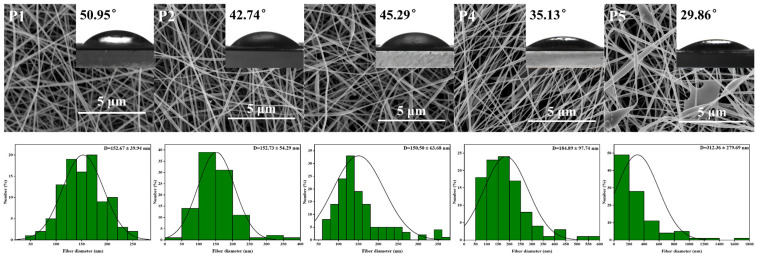
SEM images, LPFs diameter distributions, and water contact angle of P1, P2, P3, P4, and P5 nanofibers without TPs.

**Figure 4 foods-13-00284-f004:**
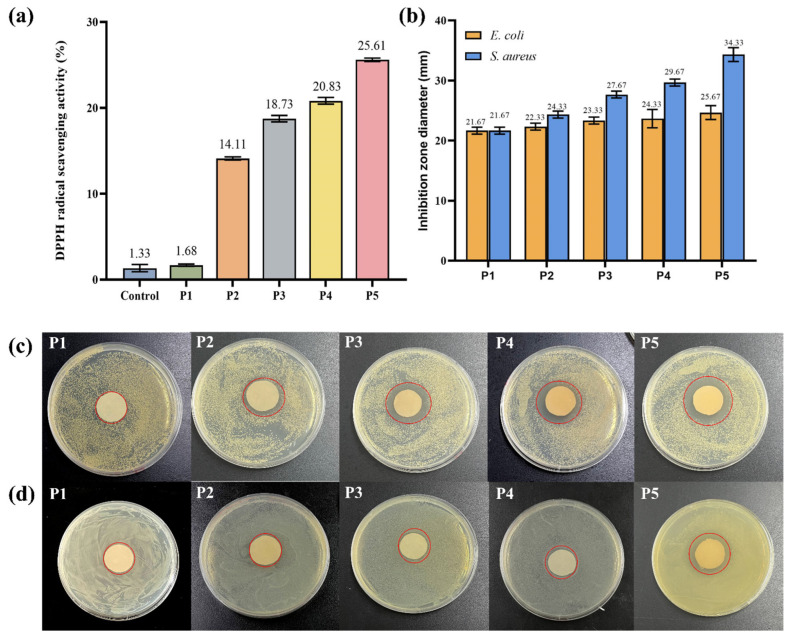
(**a**) DPPH scavenging activity of P1 (Control), P2, P3, P4, and P5; (**b**) Inhibition zone diameter of P1, P2, P3, P4, and P5 spinning solution (mm); Antibacterial effect of P1, P2, P3, P4, and P5 spinning solution on (**c**) *S. aureus* and (**d**) *E. coli*.

**Figure 5 foods-13-00284-f005:**
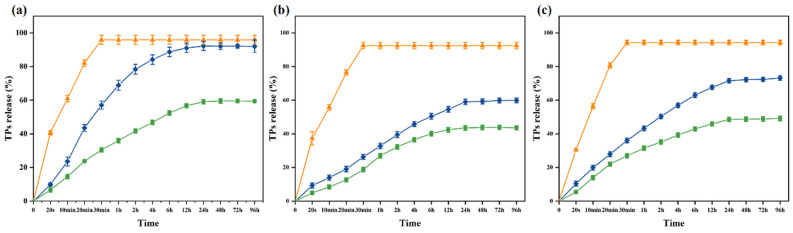
Release of TPs from nanofiber films in (**a**) 10% ethanol; (**b**) 95% ethanol; (**c**) 3% acetic acid (orange lines: TPs; blue lines: PVA nanofiber films loaded with TPs; green lines: P3).

**Figure 6 foods-13-00284-f006:**
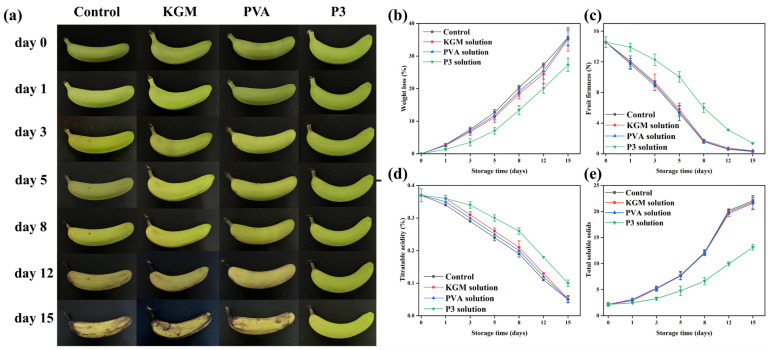
(**a**) Physical appearance, (**b**) Weight loss, (**c**) Firmness, (**d**) Titratable acidity, and (**e**) Total soluble solids of bananas with and without films during 15-day preservation (uncovered, covered with KGM, PVA, and P3).

**Table 1 foods-13-00284-t001:** Mechanical and thermal properties of P1, P2, P3, P4, and P5.

Sample Code	T_max_ (°C)	T_−5%_ (°C)	TS (MPa)	EB (%)
P1	288.21	248.81	5.40 ± 0.33 ^d^	7.24 ± 0.32 ^d^
P2	276.81	241.81	6.74 ± 0.17 ^c^	11.08 ± 0.49 ^c^
P3	281.31	231.31	10.62 ± 0.34 ^a^	18.10 ± 0.91 ^a^
P4	285.41	239.41	8.82 ± 0.42 ^b^	14.08 ± 0.43 ^b^
P5	278.01	239.41	8.66 ± 0.24 ^b^	11.34 ± 0.26 ^c^

Data are expressed as the mean ± SD (*n* = 5), different superscript letters in the same column indicate that values are significantly different (*p* < 0.05).

**Table 2 foods-13-00284-t002:** Water content (WC), swelling degree, water vapor permeability (WVP), and oxygen permeability (OP) of P1, P2, P3, P4, and P5.

Sample Code	WC (%)	Swelling Degree (%)	WVP (g mm/m^2^ kPa)	OP (g/m s Pa)
P1	3.98 ± 0.37 ^d^	318 ± 8.68 ^a^	3.81 ± 0.08 ^d^	3.68 ± 0.15 ^a^
P2	5.30 ± 0.12 ^c^	302 ± 8.23 ^b,c^	4.17 ± 0.05 ^d^	3.24 ± 0.53 ^a,b^
P3	6.37 ± 0.41 ^b^	298 ± 7.26 ^c^	6.44 ± 0.21 ^c^	2.72 ± 0.30 ^a,b^
P4	7.04 ± 0.13 ^a,b^	301 ± 4.50 ^b,c^	8.92 ± 0.37 ^b^	2.81 ± 0.28 ^b^
P5	7.76 ± 0.32 ^a^	311 ± 7.81 ^a,b^	9.61 ± 0.18 ^a^	3.00 ± 0.18 ^b^

Data are expressed as the mean ± SD (*n* = 5), different superscript letters in the same column indicate that values are significantly different (*p* < 0.05).

## Data Availability

Data is contained within the article.

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
