# Peer review of "Electrospun Konjac Glucomannan/Polyvinyl Alcohol Long Polymeric Filaments Incorporated with Tea Polyphenols for Food Preservations"

_foods, 2024, doi:10.3390/foods13020284_

Round 1

Reviewer 1 Report

Comments and Suggestions for Authors

This paper addresses the improvement of physicochemicql, antibacterial and antioxidative properties of PVA/KGM nanofibers by incorporation of tea polyphenols. It is very promissing manuscript, however I struggled to understand some parts of manuscrit:

Introduction -line from 61 to 95 need rewritting

Line 264-what does mean the best morphology?

Line from 307 to 314 need rewritting 

And finally the authors should present how nanofiber films looks like, since it is difficult to see from picture 6,  how bananas were covered with these films.

Author Response

Responses to Reviewer #1

Comments 1: Introduction -line from 61 to 95 need rewriting

Response: Thank you very much for your valuable comment. We have rewritten the section, which can be found in Line 75-95, Page 2 - 3.

Comments 2: Line 264-what does mean the best morphology?

Response: Thank you very much for your valuable comment. Uniform LPFs with an average diameter of (152.67 ± 39.94 nm) was obtained via electrospinning of PVA solutions. When adding KGM solution, the morphology of ultrafine LPFs slightly changed, and the average diameters of P2, P3, P4, and P5 LPF were 152 ± 54.29 nm, 150.50 ± 63.68 nm, 184.49 ± 97.74 nm and 312.36 ± 279.69 nm, respectively. As can be seen in Fig. 2, when the content of KGM is 0.5 % (P2), the content of KGM was too small and there was no obvious presence of KGM in the SEM image. However, when the content of KGM was too much (P4 and P5), the nanofiber membrane produced rough surface morphology and mutual adhesion. Therefore, considering the average fiber diameter and surface morphology, P3 had the best morphology.

Comments 3: Lines from 307 to 314 need rewriting

Response: Thank you very much for your valuable comment. We have rewritten line 305 to 311, which can be found on Page 8.

Comments 4: And finally, the authors should present how nanofiber films looks like, since it is difficult to see from picture 6, how bananas were covered with these films.

Response: Thank you very much for your valuable comment. During the banana preservation process, the raw banana was placed in a transparent acrylic box and the nanofiber film was covered on top of the box. When it was time for the test, the bananas were taken out onto a black background panel for photos. Immediately after the photo was taken, the banana was placed back in the box covered with the nanofiber film.

Thank you again for all your suggestions. We hope that all these changes fulfill the requirements to make the manuscript acceptable for publication in foods and I am looking forward to hearing from you soon.

Thank you and best regards.

Sincerely,

Ruojun Mu

College of Food Science

Fujian Agriculture and Forestry University

Shangxiadian Road 15#, Cangshan District, Fuzhou, 350002

E-mail: mu@fafu.edu.cn

Reviewer 2 Report

Comments and Suggestions for Authors

Natural polymers have unique advantages such as biocompatibility, biodegradability, chemical stability, and non-toxicity, but poor mechanical properties limit their applications. The authors of this paper have comprehensively described the use of KGM and PVA to prepare better quality films by electrospinning technique. Moreover, tea polyphenols were added to the film, which can be used to preserve raw bananas. Overall, the paper is well written but some issues should be properly addressed. I would recommend major revisions.

1.         Line no.23, “row banana” corrected into “raw banana”

2.         It is recommended that the experimental methodology for banana preservation should be described in detail.

3.         It is recommended that the authors add information about the relationship between specific values of TS, EB, WC, WVP, WCA, and the quality of the film to make it easier for readers to understand.

4.         Why choose 0 %, 0.5 %, 1 %, 2 %, 3 % of KGM-TPs?

5.         Line no. 205, please correct the typo errors. Moreover, did not explain how the TPs were quantified.

6.         The characters in Figures 2 and 3d are too small to read.

7.         FTIR spectrum for TPs should be added.

8.         The data described in 3.4 are not reflected in the body of the article, please confirm whether the data described in this section are missing.

9.         What are the reasons for choosing P3 out of P1-P5 for the release of TPs and experiment preservation of bananas? Please add the reasons.

10.     Please add the reasons for choosing 10 % ethanol, 95 % ethanol, and 3 % acetic acid as the three conditions for releasing tea polyphenols.

11.     It is recommended to supplement the data on indicators such as the mechanical properties of pure KGM membranes as well as antioxidant and antibacterial activities.

12.     Please refer to the following papers and update the introduction and discussion sections. https://doi.org/10.1016/j.porgcoat.2020.106010; https://doi.org/10.1016/j.ijbiomac.2023.124732

Comments on the Quality of English Language

-

Author Response

Responses to Reviewer #2

Comments 1: Line no.23, “row banana” corrected into “raw banana”

Response: Thank you very much for your valuable comment. We have corrected “row banana” into “raw banana”, which can be found in Line 23, Page 1.

Comments 2: It is recommended that the experimental methodology for banana preservation should be described in detail.

Response: Thank you very much for your valuable comment. According to your suggestion, we have made a detailed description of the experimental method of banana preservation. This section can be found in Line 217, Page 6.

Comments 3: It is recommended that the authors add information about the relationship between specific values of TS, EB, WC, WVP, WCA, and the quality of the film to make it easier for readers to understand.

Response: Thank you very much for your valuable comment. We have added the information about the relationship between specific values of TS, EB, WC, WVP, and WCA, which can be found in Line 318 - 374, Page 9 - 11.

Comments 4: Why choose 0 %, 0.5 %, 1 %, 2 %, 3 % of KGM-TPs?

Response: Thank you very much for your valuable comment. After our extensive preliminary experiments, we chose 0 %, 0.5 %, 1 %, 2 %, and 3 % KGM-TPs solution. We determined that the above five contents of KGM-TPs can be prepared by electrospinning technology. When the content of KGM-TPs in the spinning solution exceeds 3 %, it is impossible to prepare a good nanofiber film. Therefore, we finally selected 0 %, 0.5 %, 1 %, 2 %, and 3 % of KGM-TPs as the added content.

Comments 5: Line no. 205, please correct the typo errors. Moreover, did not explain how the TPs were quantified.

Response: Thank you very much for your valuable comment. We have corrected the typo errors and explained how the TPs were quantified, which can be found in Line 199 - 201, Page 5 and Line 203 – 206, Page 5.

Comments 6: The characters in Figures 2 and 3d are too small to read.

Response: Thank you very much for your valuable comment. We are sorry that the picture is too small to read. We have zoomed in on Fig. 2 and 3, which can be found in Page 7 and 9.

Comments 7: FTIR spectrum for TPs should be added.

Response: Thank you very much for your valuable comment. We used nanofiber films without TPs to determine the FT-IR, which was explained in our manuscript. This explanation can be found in Line 313, Page 9.

Comments 8: The data described in 3.4 are not reflected in the body of the article, please confirm whether the data described in this section are missing.

Response: Thank you very much for your valuable comment. We are sorry for the mistakes. We have added the data in this section, which can be found on Page 10 - 11.

Comments 9: What are the reasons for choosing P3 out of P1-P5 for the release of TPs and experiment preservation of bananas? Please add the reasons.

Response: Thank you very much for your valuable comment. The physical, mechanical, chemical, and thermal properties (e.g., morphology, TS, EB, OP, and Swelling degree) of P3 nanofiber films were the best. Therefore, P3 was selected to study the slow-release effect of TPs in food simulants and its potential in banana preservation. According to your suggestion, we have added the reasons for choosing P3.

Comments 10: Please add the reasons for choosing 10 % ethanol, 95 % ethanol, and 3 % acetic acid as the three conditions for releasing tea polyphenols.

Response: Thank you very much for your valuable comment. The migration of tea polyphenols in antibacterial packaging film is related to the type and nature of food. However, food is a complex system, so it is difficult to study the migration of tea polyphenols in food. Therefore, food simulants are usually used to study the release and migration of tea polyphenols in the world instead of various foods. According to the Chinese standard GB/T 23296.1-2009, it is stipulated that water-based food simulants include water, 3% acetic acid, 10% ethanol, and lipid food simulants include olive oil or 95% ethanol. Therefore, to observe the release characteristics of TPs from nanofiber films, three kinds of food simulants (95 % ethanol, 10 % ethanol, and 3 % acetic acid) were selected, representing fatty food, alcoholic food, and acid food, respectively. Therefore, we have chosen 10 % ethanol, 95 % ethanol, and 3 % acetic acid as the three conditions for releasing tea polyphenols.

Comments 11: It is recommended to supplement the data on indicators such as the mechanical properties of pure KGM membranes as well as antioxidant and antibacterial activities.

Response: Thank you very much for your valuable comment. Pure KGM is almost impossible to spin. The morphology of pure KGM film prepared by electrospinning was similar to that of coating. The properties of pure KGM films (such as mechanical properties, antimicrobial properties, and antioxidant properties) were very unsatisfactory. According to your suggestion, we had also tested mechanical properties, antibacterial properties, and antioxidant properties. As can be seen from Fig. S1 and Tab.3, TS and EB of pure KGM films were 4.76 ± 0.32, 7.11 ± 0.81, and the DPPH free radical scavenging rate was 10.36 %. As can be seen from Fig S1, KGM film had no inhibitory effect on E. coli and S. aureus. The properties of pure KGM films were very unsatisfactory.

Tab. 1 Mechanical and thermal properties of KGM films

Sample

code

TS

(MPa)

EB

(%)

DPPH scavenging activity (%)

Inhibition zone diameter

S. aureus

E. coli

KGM

4.76±0.32

7.11±0.81

10.67 %

N/A

N/A

Fig. S1 Antibacterial effect of KGM on (a) S. aureus and (b) E. coli. (Fig. S1 cannot be uploaded to the author's notes for reviewers, but Fig. S1 exists in the Word file of Revision letter)

Comments 12: Please refer to the following papers and update the introduction and discussion sections. https://doi.org/10.1016/j.porgcoat.2020.106010; https://doi.org/10.1016/j.ijbiomac.2023.124732

Response: Thank you very much for your valuable comment. According to your comments, we have carefully studied the two articles you sent. We have updated the introduction and discussion sections. The updated part is marked in “Red” in our manuscript. I hope our manuscript can meet your satisfactions.

Thank you again for all your suggestions. We hope that all these changes fulfill the requirements to make the manuscript acceptable for publication in foods and I am looking forward to hearing from you soon.

Thank you and best regards.

Sincerely,

Ruojun Mu

College of Food Science

Fujian Agriculture and Forestry University

Shangxiadian Road 15#, Cangshan District, Fuzhou, 350002

E-mail: mu@fafu.edu.cn

Round 2

Reviewer 1 Report

Comments and Suggestions for Authors

The authors have improved their paper, in reaction to the comments I made. Some points remain. In manuscript they should present how the acrylic box looks like, and how film looks like.

Author Response

Responses to Reviewer #1

Comments 1: The authors have improved their paper, in reaction to the comments I made. Some points remain. In manuscript they should present how the acrylic box looks like, and how film looks like.

Response: Thank you very much for your valuable comment. Based on your suggestion, we have shown in our manuscript what the acrylic box looks like, as well as what the film looks like. This section can be found in Fig. 1, Page 7. We hope that this modification will meet the needs of foods.

Thank you again for all your suggestions. We hope that all these changes fulfill the requirements to make the manuscript acceptable for publication in foods and I am looking forward to hearing from you soon.

Thank you and best regards.

Sincerely,

Ruojun Mu

College of Food Science

Fujian Agriculture and Forestry University

Shangxiadian Road 15#, Cangshan District, Fuzhou, 350002

E-mail: mu@fafu.edu.cn

Reviewer 2 Report

Comments and Suggestions for Authors

Although I received the revised manuscript, it still needs some corrections.

1. The method sections do not clearly mention how the nanofiber is utilized for the preservation of raw bananas.

2. Fig. 6. The bananas are different in the P3 group.

3. Moreover, the suggested reference is not included appropriately. For example, https://doi.org/10.1016/j.porgcoat.2020.106010.

Author Response

Responses to Reviewer #2

Comments 1: The method sections do not clearly mention how the nanofiber is utilized for the preservation of raw bananas.

Response: Thank you very much for your valuable comment. We are very sorry for neglecting this part. In our manuscript, we have added methods for preserving raw bananas using nanofibers. This part can be found in Line 217 - 221, Page 6.

Comments 2: The bananas are different in the P3 group.

Response: Thank you very much for your valuable comment. We're sorry for the confusion. In the banana preservation experiment of Group P3, we all used the same banana (as shown in Fig.S2). The size of the banana varies due to the angle and distance at which the photo is taken. We have resized the photos to make them all look uniform. These photos can be found in Fig. 6, Page 6.

Fig. S2 Physical appearance of P3,it can be seen in word document of foods-Revision letter(Reviewer#2).

Comments 3: Moreover, the suggested reference is not included appropriately. For example, https://doi.org/10.1016/j.porgcoat.2020.106010.

Response: Thank you very much for your valuable comment. We studied this article carefully and found it helpful to us a lot. Finally, we made some changes in our manuscript based on this article, which can be found in Line 33, Page 1.

Thank you again for all your suggestions. We hope that all these changes fulfill the requirements to make the manuscript acceptable for publication in foods and I am looking forward to hearing from you soon.

Thank you and best regards.

Sincerely,

Ruojun Mu

College of Food Science

Fujian Agriculture and Forestry University

Shangxiadian Road 15#, Cangshan District, Fuzhou, 350002

E-mail: mu@fafu.edu.cn
